# Biological and Medical Aspects Related to South American Rattlesnake *Crotalus durissus* (Linnaeus, 1758): A View from Colombia

**DOI:** 10.3390/toxins14120875

**Published:** 2022-12-15

**Authors:** Carlos A. Cañas

**Affiliations:** 1CIRAT: Centro de Investigación en Reumatología, Autoinmunidad y Medicina Traslacional, Universidad Icesi, Cali 760031, Colombia; cacanas@icesi.edu.co; 2Fundación Valle del Lili, Unidad de Reumatología, Cali 760032, Colombia

**Keywords:** envenomation, snake venom, snake antivenoms, crotoxin, crotamine

## Abstract

In Colombia, South America, there is a subspecies of the South American rattlesnake *Crotalus durissus*, *C. d. cumanensis*, a snake of the Viperidae family, whose presence has been reduced due to the destruction of its habitat. It is an enigmatic snake from the group of pit vipers, venomous, with large articulated front fangs, special designs on its body, and a characteristic rattle on its tail. Unlike in Brazil, the occurrence of human envenomation by *C. durisus* in Colombia is very rare and contributes to less than 1% of envenomation caused by snakes. Its venom is a complex cocktail of proteins with different biological effects, which evolved with the purpose of paralyzing the prey, killing it, and starting its digestive process, as well as having defense functions. When its venom is injected into humans as the result of a bite, the victim presents with both local tissue damage and with systemic involvement, including a diverse degree of neurotoxic, myotoxic, nephrotoxic, and coagulopathic effects, among others. Its biological effects are being studied for use in human health, including the possible development of analgesic, muscle relaxant, anti-inflammatory, immunosuppressive, anti-infection, and antineoplastic drugs. Several groups of researchers in Brazil are very active in their contributions in this regard. In this work, a review is made of the most relevant biological and medical aspects related to the South American rattlesnake and of what may be of importance for a better understanding of the snake *C. d. cumanensis*, present in Colombia and Venezuela.

## 1. Introduction

In the present review, various biological and medical aspects related to *C. durissus* and its subspecies are described: its taxonomy, anatomy, physiology, evolution, epidemiology, and clinical aspects of envenomation in humans. This includes additional descriptions of the main components of its venom and the possible development of drugs based on knowledge of its biological effects.

## 2. Overview of *C. durissus*

The description and scientific classification of the snakes of the genus *Crotalus* began to be developed in the eighteenth century [1]. Carl von Linnaeus in 1758 included snakes in the genus *Crotalus* that had venomous fangs, widened ventral scales, small subcaudals, and a terminal rattle or “crepitaculum” and identified three species: *C. horridus*, *C. dryinas*, and *C. durissus*. Linnaeus included bibliographical references from authors contemporary to him. A very interesting narration was made by Arnout Vosmaer (1767–1768), curator of the zoological cabinet of Stadholter William V of Holland in The Hague in his book “Description D’Un Serpent À Sonnette De L’Amerique” of 1767, in which a beautiful engraving of this enigmatic animal is appreciated [2]. An extensive bibliographic review of the taxonomic changes of *Crotalus* has been made [3].

The genus *Crotalus* includes 30 species, with approximately 70 subspecies [4]. It is geographically distributed from Canada to northern Argentina [5]. The species *C. durissus* has its greatest geographical distribution from Colombia to Argentina, discontinuously (in much of the Amazon, it is not reported); it has been described in some Caribbean islands, such as Aruba, and is not found in Ecuador or Chile. Eleven subspecies have been described: *C. d. durissus*, *C. d. cascavella*, *C. d. collilineatus*, *C. d. cumanensis*, *C. d. marajoensis*, *C. d. maricelae*, *C. d. ruruima*, *C. d. terrificus*, *C. d. trigonicus*, *C. d. unicolor*, and *C. d. vegrandis* [6]. The only subspecies in Colombia is *C. d. cumanensis*, found in the Caribbean, Llanos Orientales, and Magdalena Medio regions [7]. Table 1 shows the geographical locations of some of these subspecies.

The *C. durisuss* snake has two large internasal scales, four to six prefrontal scales, two to five intersupraocular scales, one to eight (usually one or two) loreal scales on each side, 11 to 18 (usually 13 to 16) supralabial scales, and between 25 and 33 dorsal scales in the medial part of the body. There are 155 to 179 scales in the ventral region in males and 163 to 190 in females, and 25 to 32 in the subcaudal region in males and 18 to 26 in females, showing sexually dimorphism [4]. They have articulated anterior fangs, which measure approximately 1 cm (Figure 1A) [17].

It has a length of approximately 100 cm but can reach up to 180 cm. Its body is bulky in the center and thinner toward the ends (Figure 1B), and at the end of the tail, it has a rattle (Figure 1C) [18]. It is very variable in the color patterns of its scales, with a mixture of gray, brown, pale red, and even almost black scales. They present with between 18 and 32 rhomboid or diamond-shaped blotches, which are very conspicuous and of high contrast in juvenile individuals (Figure 1D,E). On the head, they have two dark lines that begin in the supraocular scales and extend toward the anterior part of the trunk, with a small line that branches toward the supralabial region (Figure 1F) [4].

It is a snake of terrestrial habits. Its natural habitat is dry, sandy, or rocky areas, scattered forests, and thickets, located in ecosystems of valleys and mountain bases, generally at latitudes below 1000 m above sea level [19]. It is not very aggressive and spends most of the day resting, with more activity at night. The South American rattlesnake is active throughout the year, even during cooler and drier seasons, conditions that determine the motivation for the colonization of various geographical areas [20]. It feeds mainly on rodents and birds [21].

The origin of the snakes of the genus *Crotalus* is not yet well defined [22,23,24,25,26]. The oldest fossils of snakes of the genus *Crotalus* are from the middle Miocene (10–8 million years), but it is suggested that the true origins may be much earlier [27]. Based on the study of mitochondrial genes of *C. durissus*, a consistent phylogeographic pattern is proposed with a gradual dispersal along the Central American isthmus, followed by a more rapid dispersal through South America toward the Middle Pleistocene, approximately 1.1 million years ago [28].

## 3. Epidemiology of *C. durissus* Envenomation

Snake envenomation is the result of snakebite and venom injection, usually in accidental circumstances [29]. Approximately 5.4 million snakebites to humans occur each year, of which between 1.8 and 2.7 million cases result in envenomation and approximately 100,000 in deaths [30]. Many survivors suffer from amputations and other types of permanent disabilities. In America, approximately 100,000 snakebites and approximately 2300 deaths are reported annually [31]. In Brazil, venomous snakes are widely distributed in its territory, and occurrences show higher lethality in areas with limited access to medical care. Between 2000 and 2017, there were 471,801 cases of snake envenomation, and 1892 deaths were reported. The highest incidence occurred in the north, where 142,230 cases and 647 deaths were reported [32].

Based on Brazilian epidemiological data (Sistema de Informações de Agravos de Notificação- SINAN, 2018), in the last decade, the number of cases ranged between 26,000 and 30,000 per year, and envenomation caused by the genus *Crotalus* varied between 1700 and 2400 registered cases per year [33]. In another study carried out between 2010 and 2015 in the Brazilian Amazon, where 24 million people live, there were 70,816 snakebites, and 3058 (4.3%) cases were classified as caused by *Crotalus*, with an average incidence rate of 11.1/100,000 inhabitants/year [34].

In Colombia, between 3000 and 5000 cases of snake envenomation are recorded per year, with a mortality of approximately 300 people. Envenomation is caused in 99% of cases by snakes derived from the *Bothrops*, *Bothriechis*, *Bothrocophias*, *Porthidium*, *Lachesis* and *Micrurus* genera [35]. *Crotalus* envenomation is very rare, accounting for less than 1% of snake envenomation [36].

## 4. C. *durissus* Venom

In South America, snake venoms belonging to the *C. durissus* complex have components such as phospholipase A2 (PLA2) (including Crotoxin-CTX), snake venom serine protease (SVSP) (including collinein-1 and gyroxin), snake venom C-type lectin-like (SVCTL) (including convulxin), snake venom metalloprotease (SVMP), crotamine, snake venom L-amino acid oxidase (SVLAAO), bradykinin-enhancing peptide (BPP), crotalphine, crotalisidin, natriuretic peptide isolated from *C. d. cascavella* (NPCdc), hyaluronidase, phosphodiesterase from *C. d. collilineatus* (CdcPDE), nucleotidases, and snake venom vascular endothelial growth factor VEGF (SVVEGF) [37,38,39,40,41,42,43,44], responsible for the biological effects and aspects related to envenomation. The relative concentrations and interactions between venom components such as ubiquity, bioactivity, and number of associations and synergies is a very important area to be explored in *C. durissus* venom [45].

Intragenus and intraspecific variation in venom in pit vipers has been correlated to diet or topographic features. One of the primary reasons for the high diversity and plasticity in snake venom is frequent duplication of toxin-encoding genes and recruitment strategies [45]. Variability has been documented in the percentages of the different components of the venom, based mainly on proteomic studies (see Table 1). This may be the cause of variations in the effect of envenomation and response to antivenom [46,47,48]. In the case of *C. durissus*, there are differences mainly in PLA2 and crotamine [49,50,51,52,53,54,55,56,57,58,59,60,61].

### 4.1. PLA2—CTX

CTX is the main component of the venom of *C. durissus* [62]. It is characterized by being a protein complex composed of two noncovalent subunits, the basic PLA2 subunit (CB) and the acid subunit, crotapotin or PLA2 Asp49 (CA) [63,64]. The two subunits act in synergy. CA is a chaperone that assists the binding of CB to binding sites on nerve endings and skeletal muscle [65,66]. The action on the nerve endings is of the beta-neurotoxin type, preventing the release of acetylcholine from the neuronal component of the neuromuscular plate, which causes its paralytic action [67,68,69,70]. This mechanism of action has been proposed as the basis for the development of muscle relaxants [71], with a possible action and response similar to that of botulinum toxin type A [72].

A postsynaptic effect at the level of the acetylcholine receptor in the muscular component of the neuromuscular junction has also been postulated, with an action similar to that of α-neurotoxins, an effect that would contribute to the paralysis of its prey [73].

In the venom of each individual *C. durissus,* there is a mixture of isoforms of CA (CA1, CA2, CA3, CA4) and CB (CBa, CBb, CBc, CBd) [74,75], which may have different biological properties [76]. Based on these differences, two classes of CTX (classes I and II) are distinguished. Class I, composed of CBb, CBc, and CBd with any of the CA isoforms, forms very stable complexes (half-life 10–20 min, Kd = 4.5 nM) with high toxicity and low PLA2 activity. Class II, composed of CBa2 with any of the CA isoforms, contains fewer stable complexes (half-life of approximately 1 min, Kd = 25 nM) with high enzymatic activity [77]. The three-dimensional structure of the molecule and its subunits has been extensively studied [78,79].

CTX-related myotoxicity is characterized by sarcolemma degradation by phospholipid hydrolysis [80,81]. The release of mitochondrial “alarmins” secondary to myonecrosis could contribute to the local and systemic inflammatory events observed in *C. durissus* envenomation [82]. Altered intracytoplasmic calcium dynamics are also implicated in muscle damage [83].

A possible direct neuronotoxic effect of CTX, which can lower the seizure threshold, has been studied [84]. CTX induces calcium-dependent glutamate release through the N and P/Q calcium channels, which suggests a role in this effect [85]. The activation of alpha-adrenergic receptors and 5-hydroxytryptamine (5-HT) has been shown, which can cause an antinociceptive effect, with potential research importance as an analgesic [86].

CTX has an anticoagulant effect by decreasing the levels of von Willebrand factor (vWF) and tissue plasminogen activator (t-PA) and raising the levels of protein C and plasminogen activator inhibitor type 1 (PAI-1), an effect that has been studied as a potential tool for the development of antithrombotic drugs [87].

In the search for biomedical applications, there are different lines of research aimed at evaluating the effect of CTX on the immune system (aiming at possible development of anti-inflammatory and immunosuppressive drugs) [88], as well as its possible anti-infection and anti-neoplastic action.

Inhibitory actions of CTX on dendritic cells [89,90], neutrophils [91,92], monocytes/macrophages [93,94], and T and B lymphocytes [95,96] have been studied. In macrophages, the effect of CTX on phagocytic propagation and activity has been characterized, and toxin-mediated alterations in cytoskeletal proteins modulate phagocytosis independent of the receptor involved [97]. The CB fraction of CTX can decrease the expression of MHC type II molecules, which are important for antigen presentation to T-lymphocytes, as well as costimulatory molecules such as CD40, CD80, and CD86 [92]. It has been shown to inhibit antibody production [98], a concern in the manufacture of antivenoms [99,100]. An inhibitory effect on proinflammatory prostaglandins has also been reported [101,102,103,104]. In an experimental model of sepsis, CTX improved the survival of mice from 40% to 80%, modulating the secretion of proinflammatory cytokines such as IL-6 and TNF-α while increasing IL-10, a known inhibitor of the immune response [105]. In models of autoimmune encephalomyelitis, it had an anti-inflammatory and immunosuppressive effect [106,107], as well as in a model of inflammatory bowel disease [108].

CTX as an anti-infection agent has been studied as an antiviral, with action on hepatitis C [109], dengue [110], yellow fever [111], and chikungunya [112] viruses; as an antibacterial [113]; as an antifungal [114]; and as an antiplasmodium [115].

CTX as an antineoplastic agent has been studied in various neoplastic cell lines, such as those of the thymus [116], lung [117,118], breast [119], pancreas, esophagus, glial cells, and cervix [120].

CTX increases the cystic fibrosis transmembrane conductance regulator (CFTR) chloride channel current and corrects ∆F508CFTR dysfunction, which could have an impact on cystic fibrosis [121].

### 4.2. SVSP—Collineína-1, Gyroxin

SVSPs, present in many viper venoms, have a thombin-like effect [122], generating fibrin from fibrinogen. In *C. d. collilineatus*, one of these proteins has been isolated, collinein-1, which has been extensively studied from the molecular point of view, as well as in terms of its biological effects [123]. A similar enzyme has been isolated from *C. d. terrificus* [124]. Due to its action of generating fibrin, it has been studied in wound healing [125,126], tendon plasty [127], cartilage repair [128], motor neurons [129], and scaffolding for mesenchymal cells [130]. It is proposed that pegylation of these SVSPs can enhance their therapeutic efficiency [131].

An inhibitory effect of collinein-1 on the cancer-relevant voltage-gated potassium channel (hEAG1) has been reported [132].

Gyroxin is another SVSP found in the venom of *C. d. terrificus* and causes “gyroxin syndrome” in mice, characterized by aberrant motor events, known as barrel rotation [133]. It possibly induces neurotoxicity by increasing glutamate levels [134] and by altering blood-brain barrier permeability [135]. Similar to other SVSPs, gyroxin promotes fibrinogen fractionation into fibrin monomers, resulting in thrombus formation [136].

Gyroxin has a proinflammatory effect that is involved in the degradation of protease activated receptors PAR1 and PAR2, which activate phospholipase C (PLC) and protein kinase C (PKC) [137]. The edema-inducing effect of gyroxins is under investigation; however, effects on prostaglandins have been implicated [138].

### 4.3. SVCTL—Convulxin

Convulxin has been isolated from the venom of *C. d. terrificus*. It is an octamer with disulfide bonds composed of four alpha subunits and four homologous beta subunits, having similarities with CTLs, factor IX-binding protein (IX-bp) and flavocetin-A (Fl-A) [135]. Convulxin differs from these proteins in that it lacks the consensus sequence for carbohydrate and Ca2+ binding [139,140]. It produces the activation of platelet aggregation through an agonist effect on the receptor for glycoprotein VI (GPVI), which occurs in exposed collagen when there is endothelial damage [141]. It causes a form of neurotoxicity with balance disturbances and seizures, best observed in mice [142].

Little is known about the effect of convulxin on immune cells. A study demonstrated its role in inflammasome activation and increased IL-1 release [143].

Convulxin corresponds to 0.04% of the protein components of *C. d. terrificus* venom [13].

### 4.4. SVMP

SVMPs constitute a large family of biologically active proteins isolated from the venom of various species of the Viperidae family [144]. SVMPs comprise zinc-dependent enzymes of the reprolysin family. The members of this group of metalloproteases are classified into four main classes (I, II, III and IV) according to the presence of different domains: class I (20–30 kDa) includes enzymes that have only one metalloproteinase domain; class II (30–60 kDa) includes proteins with a metalloproteinase domain and a disintegrin-like domain (not enzyme); class III (60–100 kDa) includes the two domains of class II plus a cysteine-rich domain; and class IV includes proteins with all the aforementioned domains linked to a C-type lectin subunit by disulfide bonds [145]. SVMPs induce bleeding disorders, tissue necrosis including skin necrosis [146], myotoxic effects, inflammatory reactions, and endothelial cell injury [147,148]. In the venoms of the subspecies of *C. durissus,* the presence of the four classes of SVMPs has been reported [12].

SVMPs with a disintegrin domain (SVMP class II) can release this domain, which has mainly RGD (arginine-glycine-aspartic acid) motifs and less common “non-RGD” motifs, such as lysine-tryptophan-serine (KTS), arginine-tryptophan-serine (RTS), and methionine-leucine-aspartic acid (MLD) motifs, among others, with the ability to bind to integrins [149,150,151]. Disintegrins separate cells from the extracellular matrix; in the case of the skin, this is expressed clinically as blisters, as seen in *Bothrops* snake envenomation [152]. It has been postulated that this biological mechanism may be useful for the development of antimetastatic drugs [153].

### 4.5. Crotamine

Crotamine is a small basic protein with a size of 4.8 kDa and an isoelectric point of approximately 10.8 [154,155,156]. It is composed of a single chain of 42 amino acid residues and contains three disulfide bonds [157,158]. The general folding of crotamine is homologous to antimicrobial peptides belonging to the families of alpha-defensins, beta-defensins, and insect defensins [159,160]. This toxin is myotoxic [161], with a paralyzing effect on prey [162,163]. It is a cause of myotoxicity during envenomation in humans. It acts on sodium and potassium channels [164,165] and generates mitochondrial dysfunction [166]. It was first observed in the venom of Argentine rattlesnakes by Gonçalves and Polson and was later found in other venomous rattlesnakes from southern Brazil [167].

Crotamine has a good capacity for cell penetration [168,169], given both its small size and its positive net surface charge [170], which makes it attractive in the study of biotechnological applications. Several biological functions of this polypeptide have been described, including antiviral (SARS-CoV-2) [171], antibacterial [172], antifungal [173,174], antileishmanial [175,176,177], anthelmintic [178,179], antimalarial [180] and antitumor [181,182,183] activities.

Mice injected intradermally with crotamine exhibited acute local and systemic inflammatory responses similar to histamine, limiting the therapeutic use of crotamine in its original form [184]. It stimulates the phagocytic and cytostatic activity of macrophages by inducing NO and TNF-α through the p38 and NF-κB signaling pathways [185]. It may additionally have antinociceptive [185], activation of platelet aggregation [186] and anti-inflammatory [187] effects.

### 4.6. SVLAAO

SVLAAO is a dimeric enzyme that deaminates an L-amino acid to an α-keto acid with concomitant production of hydrogen peroxide and ammonia [188]. These enzymes are widely distributed in the venom of snakes, including *Crotalus* [189], and induce various biological effects, including apoptosis [190], edema [191], increased platelet aggregation [192,193], and bleeding disorders by factor IX inhibition [194], among others. It can have antiviral [195], antibacterial, and antifungal [196] effects. In the specific case of *C. d. cascavella* [197], proinflammatory, antibacterial, and antileishmanial effects of SVLAAO have been found. Bordonein-L from *C. d. terrificus* exhibits cytotoxicity against the fibroblast cell line and kills *Leishmania amazonensis* promastigotes [198]. SVLAAO from the venom of *C. d. cumanensis* has been found to have an antibacterial effect [199]. Other effects have been studied, such as the antineoplastic effects of SVLAAO from the venom of *C. d. terrificus* in glioma and pancreatic carcinoma cell lines [200].

### 4.7. BPP

Some snakes produce BPP in their venom, which increase the hypotensive effect induced by bradykinin [201] and decrease the vasopressor effect of angiotensin I by inhibiting angiotensin-converting enzyme (ACE) as reported by Kelvin K-C Ng and John Robert Vane in 1967 [202]. This substance was studied in the snake *B. jararaca* from Brazil. JR Vane motivated two of his students, David Cushman and Miguel Ondetti, to develop an ACE-inhibiting drug, and thus captopril was synthesized [202]. JR Vane was awarded the Nobel Prize in Physiology or Medicine in 1982 for his study of prostaglandins and the mechanisms of action of aspirin [203]. From the venom of *C. d. cascavella*, a form of BPP was isolated with a higher potency than that of *B. jararaca* [204].

### 4.8. Crotaline

Giorgi et al. (1993) [205] demonstrated that factors with molecular masses below 3 kDa present in the venom of *C. d. terrificus* cause antinociceptive effects in mice that are likely mediated by opioid receptors. Subsequently, crotalphine, a 14-amino acid peptide containing a disulfide bridge and pyroglutamic acid, was isolated [206]. It exerts a potent and long-lasting antinociceptive effect that is mediated by the activation of peripheral opioid receptors [207,208]. Crotalphine is not a direct opioid receptor agonist; however, it induces the release of dynorphin A, which activates kappa opioid receptors [209]. Opioid receptor activation regulates a variety of intracellular signals, including the mitogen-activated protein kinase (MAPK) pathway. In primary cultures of sensory neurons, crotalphin increased the levels of activated ERK1/2 and JNK-MAPK, and this increase depended on the activation of protein kinase Cζ (PKCζ). In vivo pharmacological inhibition of spinal ERK1/2 and JNK, but not p38, blocks the antinociceptive effect of crotalphin [210]. The peripheral L-arginine-nitric oxide-cyclic GMP pathway and ATP-sensitive K⁺ channels are involved in the antinociceptive effect of crotalphin on neuropathic pain in rats [211].

Crotalphine has also been reported to desensitize transient receptor potential ankyrin 1 (TRPA1) ion channels [212], a receptor that has a relevant role in the maintenance of inflammatory hyperalgesia [213]. Crotalphin attenuates pain and neuroinflammation in experimental autoimmune encephalomyelitis in mice [214].

### 4.9. Crotalicidin

Cathelicidins are antimicrobial peptides produced by humans and animals in response to various pathogenic microbes. Crotalicidin, a peptide related to cathelicidin from the venom of the rattlesnake *C. d. terrificus*, has shown antibacterial [215], antiparasitic [216], and antifungal [217,218] activity, similar to that of the human cathelicidin LL-37. Crotalicidin has also been studied for its antiproliferative properties [219,220]. It is proinflammatory [221].

### 4.10. NPCdc

NPCdc isolated from *C. d. cascavella* reduces the tubular transport of sodium, which causes an increase in its excretion and generates a diuretic action. Additionally, through a mechanism possibly mediated by nitrites, it has a vasodilator effect [222]. An antioxidant effect has been postulated [223].

### 4.11. Hyaluronidase

Hyaluronidases are a common component of snake venoms and are known as “spread factors” because they cleave hyaluronate, a nonproteoglycan polysaccharide found in the extracellular matrix, facilitating the diffusion of toxins into tissues and blood circulation of prey/victims [224,225]. Although they are not toxins, they indirectly enhance the toxicity of the venom [226], contributing to local and systemic damage. A hyaluronidase has been isolated from *C. d. territicus*, with anti-edema properties [227].

### 4.12. CdcPDE

CdcPDE has been isolated and characterized from the venom of the snake *C. d. collilineatus* and has been shown to have an inhibitory effect on platelet aggregation and cytotoxic action on human keratinocytes [228].

### 4.13. Nucleotidases

In the venom of *C. d. terrificus*, small membranous vesicles contain various bioactive molecules, among which ecto-5’-nucleotidase has been isolated. Ecto-5’-nucleotidase is a cell membrane protein that releases adenosine and generates vasodilation, paralysis, and anticoagulation [229].

### 4.14. SVVEGF

SVVEGF contributes to the action of venom components by increasing vascular permeability. It has been isolated from *C. d. collilineatus* (CdcVEGF) [230].

## 5. PLA2 Inhibitors (in Blood-Not in Venom)

Despite the deleterious action of the venom components of *C. durissus*, snakes of this species are naturally resistant to them due to the presence of specific antitoxins in their circulating blood [231]. Antitoxins are proteins secreted by the liver of the snake capable of preventing damage caused by toxins that may eventually reach the bloodstream, the most important being those that inhibit PLA2 (PLI) [232,233].

PLIs are oligomeric glycoproteins, with a molecular mass between 75 and 180 kDa, classified into three structural classes (αPLI, βPLI, γPLI). αPLIs have a C-type lectin-like domain and preferentially neutralize the acid PLA2. βPLIs are distinguished by the presence of leucine-rich repeats and are capable of inhibiting basic PLA2 [234]. γPLIs are composed of a conserved half-cysteine tandem repeat known as three-finger motifs [235]. The first endogenous PLI isolated from *C. d. terrificus* was a γPLI [236]. The molecular structure and possible interaction mechanisms between CNF and CTX have been investigated using biochemical and biophysical approaches [237]. CNF has been shown to inhibit the toxic effects of CTX in mouse neuromuscular preparations [238].

PLI has been shown to modulate human peripheral blood mononuclear cells and neutrophils, generating an anti-inflammatory action [239].

The manufacture of a recombinant PLI from *C. d. collilineatus* (recγCdcPLI) is highly efficient and therefore allows for improved drug design for the treatment of diseases caused by PLA2 activity [240]. An in vitro antitumor effect has been reported [241].

PLI in *C. d. terrificus* corresponds to 0.46% of the total venom proteins [13].

## 6. Human Envenomation by *C. durissus*

At Fundación Valle del Lili, a high-complexity university hospital located in Cali, in southwestern Colombia, during the last two decades (2001–2022), we have treated 104 patients bitten by snakes of the Viperidae family, with 84 of the cases leading to envenomation. Only two patients were bitten by *C. d. cumanensis*, one with dry bite (without envenomation), a condition that has already been reported in this type of bite [242], and one with envenomation. This case was treated in June 2004. A 35-year-old man had a pet *C. d. cuminensis* snake, which was large (a situation that may be related to the development of greater myotoxicity [243]). The animal was from the Colombian Caribbean region. He was bitten on the second finger of his left hand after improper handling of the animal. He reported mild pain and progressive development of edema that spread to the forearm (Figure 2). Four hours after the bite, he began to present blurred vision, slight weakness in the muscles of the face and the four extremities associated with generalized myalgia. He was hospitalized and found to have hypotension (blood pressure: 90/70 mmHg, pulse: 102 ppm). Physical examination revealed edema of the left upper limb, a small punctiform wound on the second finger of the left hand with no signs of bleeding, slight bilateral palpebral ptosis and decreased generalized muscle strength without respiratory distress. Laboratory tests showed mild anemia, neutrophilia, prolonged coagulation tests, slight consumption of fibrinogen, and severe increase in creatine phosphokinase (CPK) (Hb: 10.2 g/L, white blood cell count: 12,054/mm^3^, neutrophils: 9250/mm^3^, lymphocytes: 1954/mm^3^, monocytes: 552/mm^3^, platelets: 112.00/mm^3^, prothrombin time: 16 s, partial thromboplastin time: 54 s, fibrinogen: 154 mg/dL, CPK: 1174.5 mcg/L). Treatment was started in the intensive care unit (ICU), where he received 12 vials of polyvalent antivenom serum for snakes of the Viperidae family, manufactured at the Instituto Nacional de Salud, Colombia [244]. Twenty-four hours after admission, he presented with an increase in blood urea nitrogen (BUN) and creatinine (BUN: 62 mg/dL and creatinine: 3.2 mg/dL), and he became oliguric with dark urine, consistent with myoglobinuria. Acute renal failure was diagnosed, for which he was treated with several dialysis sessions. The patient gradually recovered clinically and paraclinically. The patient was discharged with no apparent sequelae.

The clinical manifestations observed in envenomation by snakes of the C. durissus species are the result of local tissue damage (edema) [245], neurotoxic activities (neuromuscular blockade) [246], myotoxicity, hematotoxicity (hemolysis) [247], nephrotoxicity, and coagulopathy [248], as seen in our patient. Early consultation is important for timely treatment, given that the manifestations can evolve into life-threatening conditions or leave serious sequelae, such as the progression of local damage or development of a compartment syndrome [249]; the evolution of neurotoxicity leading to difficulty swallowing, velopalatine paralysis, increased vomiting reflex, changes in taste and smell, and finally respiratory arrest; and myotoxic effects that lead to the devastation of the muscles and further worsening renal compromise that can lead to irreversible damage [250,251,252,253]. Coagulopathic effects can lead to afibrinogenemia with blood incoagulability [254].

Kidney damage is the product of rhabdomyolysis [255,256], hypotension [257,258], and a direct toxic effect of the venom components on the kidney [259,260,261,262]. It most certainly contributes to the cardiotoxic effect [263,264,265,266,267], which can contribute to decreased cardiac output and worsened blood perfusion [268].

Toxicity has also been described at the neuronal [84,85], pulmonary [269,270], hepatic [271,272] and germ-cell [273] levels.

Based on clinical manifestations, *C. durissus* envenomation can be classified as mild, moderate, or severe. Mild envenomation is characterized by discrete neurotoxic signs and symptoms, without myalgias or with mild myalgias and without urine discoloration; moderate envenomation is characterized by the presence of discrete neurotoxic signs and symptoms, discrete myalgias and myoglobinuria; and severe envenomation is characterized by obvious and intense neurotoxic signs and symptoms (myasthenic facies, muscle weakness), intense and generalized myalgia, dark urine, and oliguria or anuria [274].

## 7. Antivenoms Used in *C. durissus* Envenomation

### 7.1. Equine Antivenoms

The only treatment available for snakebite envenomation is antivenom, which is a hyperimmune immunoglobulin obtained from animals immunized with specific venom [275], a technique that has remained relevant since its description by Albert Calmette in 1896 [276]. At present, mixtures of venoms of different species and/or subspecies are made from different geographical areas (taking into account interspecies variations) belonging to the same genus or family [277]. The antigens are usually inoculated into horses (immunization process), followed by a screening test (approximately 15 to 30 days later) to investigate the titer of specific antibodies. If the expected antibody titers are achieved, blood is obtained from the animal, the plasma is separated, and the IgG immunoglobulins are purified, which can be prepared in three main conformations: monovalent Fab, F(ab’)2 fragments, or complete IgG [278,279,280,281]. The neutralizing capacity of antivenoms can be determined by different techniques known as “antivenomics” [282]. There are studies of the ability of antivenoms to neutralize *C. durissus* venoms [283], including interspecies [284] and intersubspecies [285].

In Colombia, one of the commercial antivenoms available is that produced by INS, which is a polyvalent heterologous serum made by inoculating horses with the venom of *Crotalus*, *Lachesis*, and *Bothrops* species. A 10 mL vial has the capacity to neutralize 10 mg of *C. durissus* venom [244]. Another antivenom available in Colombia is manufactured in Mexico by the Bioclón Institute. It is a polyvalent antiviperid (Antivipmyn-Tri^®^) digested with pepsin (fabotherapeutic type F(ab’)2) from horse plasma; 10 mL of Antivipmyn-Tri^®^ neutralizes 15 mg of *Crotalus* sp. Venom [286,287]. In one case with respiratory paralysis, it was very effective [288].

Given the political conflicts in Colombia, there is difficulty in accessing biological samples in several regions. A strategy of making biobanks of venoms and mRNA of the most relevant proteins from *C. d. cumanensis*, such as SVMP, disintegrins, disintegrin-like, PLA2, SVTCL, and SVSP, with a view to obtaining future products with an antivenom effect, has been proposed [289].

Apart from antivenom, which is the cornerstone of treatment, support measures are also fundamental for the control of cardiovascular, pulmonary, renal, and neurological effects, among others; ideally, the patient should be managed in the ICU [290].

The PLA2 inhibitor varespladib is highly effective in abrogating the neuromuscular blocking activity of *C. durissus* venoms [291,292] and appears to act synergistically with antivenom [293]. Clinical trials are not yet available.

Radicicol, a heat-shock protein (HSP) inducer, enhances muscle regeneration by attenuating NF-kB activation and increasing myogenic differentiation. It could be useful in the regeneration of skeletal muscle injured by CTX [294,295].

### 7.2. Nanobodies

Camelids produce immunoglobulin G devoid of light chains, whose recognition domain is a single-domain antibody (VHH). VHH has been obtained against CTX of *C. d. terrificus* with good neutralizing capacity and is a promising future alternative for the treatment of this type of envenomation [296,297].

## Figures and Tables

**Figure 1 toxins-14-00875-f001:**
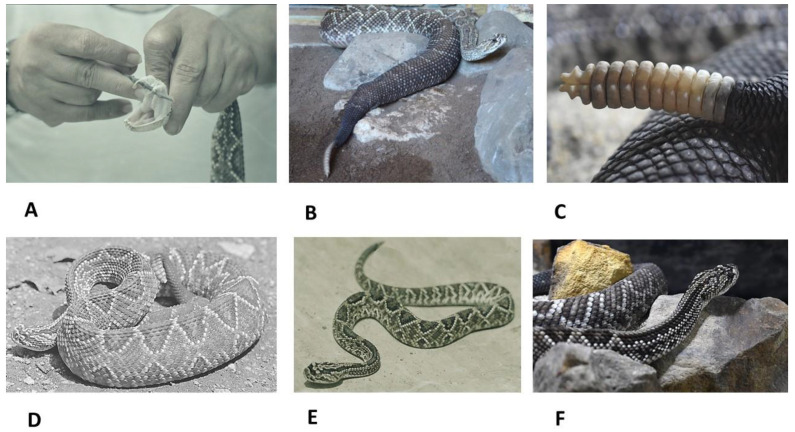
The snake *C. d. cumanensis*. (**A**): Large, articulated, anterior fangs are visible. (**B**): General appearance of the animal, where the shape of its body can be seen, wide in the central part and thin at the ends. (**C**): A typical rattle. (**D**): Method of perching in a dangerous situation. The patterning of its scales can be seen. (**E**): Young individual, in whom the body patterning is very well appreciated. (**F**): Front part of the snake showing parallel dorsal lines that extend from the head to the front part of the trunk.

**Figure 2 toxins-14-00875-f002:**
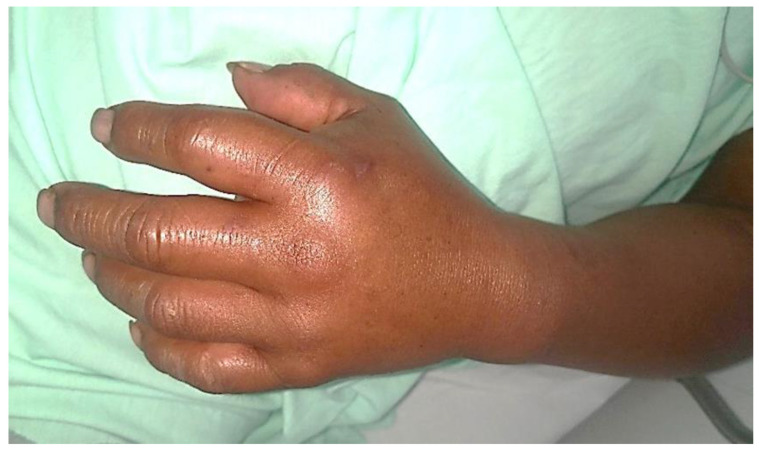
The punctiform wound caused by one of the fangs of a *C. d. cumanensis* snake, showing the development of edema in the hand in the early stage of envenomation.

**Table 1 toxins-14-00875-t001:** Most studied subspecies of *C. durissus*, indicating their geographical location and the percentages of the most relevant components of their venom.

*C. durissus*	Reported Region	Crotoxin	SVSP	CTL	SVMP				Crotamine	LAAO	BIP	Disintegrin	Others	Refs.
					I	II	III	IV						
*C. d. cumanensis* Humboldt, 1833	Colombia and Venezuela	64.71	6.33	1.18	-	-	3.3	-	0.0 *–5.77	3.16	-	13.7	1.85	[8,9,10]
*C. d. ruruima* Hoge, 1966	North of Venezuela	82.7	8.1	4.3	–	–	2.9	–	1.5	<0.5	<0.1	-		[11]
*C. d. cascavella* Wagler, 1824	North of Brazil	72.5	1.2	<0.1	–	–	<0.1	–	–	<0.1	–	-	20.3	[12]
*C. d. collilineatus* Hoge, 1966	Northeast of Brazil	67.4	1.9	<0.1	–	–	0.4	–	20.8	0.5	–	-	13.8	[12]
*C. d. terrificus* Laurenti, 1768	Centre of Brazil	48.5–82.7	0.7–25.3	<0.1–2.7	0.09–5.5	1–19	0.6–4.5	1.8	0.5–22.3	48.5–82.7	0.7–25.3	-	<0.1–2.7	[3,11,13,14,15]
*C. d. durissus* Linnaeus, 1758	South of the Amazonian forest of Brazil, extreme southeast of Peru, Bolivia, Paraguay, Uruguay, north of Argentina	68	5.1	<0.2			2.4		12	3.6	0.9	7.9		[16]

* *C. d. cumanensis* species from the Colombian Caribbean are Crotamine-negative, unlike individuals from the Orinoquía and Magdalena Medio. SVSP: snake venom serine protease, CTL: C-type lectin-like, SVMP: snake venom metalloprotease, LAAO: L-amino acid oxidase, BPP: bradykinin-enhancing peptide.

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
