# Peer review of "Biological and Medical Aspects Related to South American Rattlesnake Crotalus durissus (Linnaeus, 1758): A View from Colombia"

_toxins, 2022, doi:10.3390/toxins14120875_

Round 1

Reviewer 1 Report

Overall, I think the manuscript can be made more oriented, succinct, and structured. There is good information, but I find that the manuscript in general is missing key information that would be crucial to have well-developed arguments. The manuscript title says it’s focusing on the venom of C. durissus but the manuscript is often discussing other species. Several key studies such as Deshwal et al. (2021) Meta-Analysis of the Protein Components in Rattlesnake Venom. Toxins, 13(6), 372 are missing.

Below are a few comments

·      Line 36-37 need not be in a separate paragraph.

·      I will suggest starting the manuscript from line 38 as lines 25-37 are not adding much to the manuscript. 

·      Line 88 -I will suggest removing it.

·      I think sections- Taxonomy and Distribution can be combined with the section Overview of C. durissus and the section labeled evolutionary aspects of C. durissus and made succinct as that information does not directly align with Toxins

·      Line 148-149 – There are several factors leading to the intraspecific venom variation, please cover them all. Refer to Deshwal et al. (2021) Meta-Analysis of the Protein Components in Rattlesnake Venom. Toxins, 13(6), 372 for more details. 

·      Line 151-162 in the section venom of C. durissus is not focused on the venom profile of C. durissus.

·      Line 164 – inline citation needs to be done correctly. 

·      Line 164-166 – this information though interesting does not add much to the manuscript. 

·      There are key details missing from the manuscript such as in lines 21-216 what about inhibitory actions of CTX have been studied.

Author Response

REPLIES TO REVIEWER 1

TOXINS Reference manuscript:  Biological and medical aspects related to South American rat-tlesnake Crotalus durissus (Linnaeus, 1758): A view from Colombia.

Thank you very much for the revision of the manuscript and the pertinent suggestions to improve it.

  • Line 36-37 need not be in a separate paragraph.

Reply: Suggested change was made.

  • I will suggest starting the manuscript from line 38 as lines 25-37 are not adding much to the manuscript.

Reply: the phrases were removed

  • Line 88 -I will suggest removing it.

Reply: the phrase was removed

  • I think sections- Taxonomy and Distribution can be combined with the section Overview of C. durissus and the section labeled evolutionary aspects of C. durissus and made succinct as that information does not directly align with Toxins

Reply: Suggested change was made.

  • Line 148-149 – There are several factors leading to the intraspecific venom variation, please cover them all. Refer to Deshwal et al. (2021) Meta-Analysis of the Protein Components in Rattlesnake Venom. Toxins, 13(6), 372 for more details.

Reply: Suggested extension were made. Reference was included.

  • Line 151-162 in the section venom of C. durissus is not focused on the venom profile of C. durissus.

Reply: Suggested change was made.

  • Line 164 – inline citation needs to be done correctly.

Reply: Suggested change was made.

  • Line 164-166 – this information though interesting does not add much to the manuscript.

Reply: the phrase was removed.

  • There are key details missing from the manuscript such as in lines 21-216 what about inhibitory actions of CTX have been studied.

Reply: Suggested changes and extension were made.

Reviewer 2 Report

This review about C. durissus in particular C. d. cumanensis is well documented and very pleasant to read.

Here are some minor remarks

- line 45: write Crotalus in italics

- lines 142-143: clearly distinguish between SVMP-II having a disintegrin domain and true (or short-coding) disintegrins which are not enzymes

- line 207: antifibrinolytic rather than profibrinolytic

- the paragraph between lines 287 and 291 deals with true disintegrins and should be separated from 6.4. SVMP

- lines 497-499: specify that the efficacy of varespladib has only been demonstrated under experimental conditions and that to date there is no clinical data

Author Response

REPLIES TO REVIEWER 2

TOXINS Reference manuscript:  Biological and medical aspects related to South American rat-tlesnake Crotalus durissus (Linnaeus, 1758): A view from Colombia.

Thank you very much for the revision of the manuscript and the pertinent suggestions to improve it.

  • line 45: write Crotalus in italics

Reply: Suggested change was made.

  • lines 142-143: clearly distinguish between SVMP-II having a disintegrin domain and true (or short-coding) disintegrins which are not enzymes

Reply: the explanation was made.

  • line 207: antifibrinolytic rather than profibrinolytic

Reply: Suggested change was made.

  • the paragraph between lines 287 and 291 deals with true disintegrins and should be separated from 6.4. SVMP

Reply: Suggested change was made.

  • lines 497-499: specify that the efficacy of varespladib has only been demonstrated under experimental conditions and that to date there is no clinical data

Reply: the clarification was made.